# Emotional flexibility and general self-efficacy: A pilot training intervention study with knowledge workers

**Jacqueline Brassey**[1,2,3]*, **Arjen van Witteloostuijn**[1,4], **Csaba Huszka**[3], **Tobias Silberzahn**[5], **Nick van Dam**[2,6]

**1** Vrije Universiteit Amsterdam, Amsterdam, The Netherlands, **2** IE University, Segovia, Spain, **3** Maastricht University, Maastricht, The Netherlands, **4** Antwerp Management School / University of Antwerp, Antwerp, Belgium, **5** McKinsey & Company, Berlin, Germany, **6** Nyenrode Business University, Breukelen, The Netherlands

* Jacqueline.brassey@gmail.com

**Data Availability Statement:** All relevant data are within the manuscript and its Supporting Information files.

## Abstract

Emotional flexibility advancement has been found to be highly effective in clinical settings to treat, for example, depression, anxiety, and chronic pain. Developing these skills in the working context has also shown very encouraging results in public sector settings. Also, a few studies have revealed effectiveness in a private sector setting, but no studies have yet looked at the effectiveness of developing these skills amongst high-paced, high-demanding, and highly-educated knowledge workers. In this pilot training intervention study, we report evidence that emotional flexibility can be developed in this context. We conducted an experiment with treatment and control groups, with only the treatment group receiving an emotional flexibility training. Emotional flexibility improved significantly for the treatment group, whereas the improvements were minimal or negative for the control group. Furthermore, we reveal that General self-efficacy improved amongst treatment group participants (and not for control group participants), and that this is associated with emotional flexibility. Finally, we show that the improvements were higher for participants starting from a lower baseline.

## Introduction

The workplace environment is changing fast. The digitization and the upcoming impact of artificial intelligence ask for new ways of working, pushing reskilling needs up. Research by McKinsey & Company predicts that up to 30% of work will be potentially displaced by automation by 2030, and up to 9% of the workforce in 2030 will be in new occupations [1]. Another study predicted that almost half of the jobs in the US will be impacted by automation by 2033 [2]. The strong push for automation and the need to reskill the workforce in advanced technical skills are associated with a simultaneous push for stronger development of 'softer' competences, including emotional intelligence, communication, and creativity [3]. At the same time, according to the Gallup organization, employee engagement is at an all-time low [4]. Pfeffer goes even further by arguing that stress in the workplace is at an all-time high, being damaging to health [5].

**Funding:** This paper's study was conducted as part of the International Master in Affective Neuroscience of Maastricht University and the University of Florence. The 1st author was a student of this program at the time whilst working 80% at McKinsey & Company. She received permission to conduct the study, with help of a colleague from the German office. The study was executed during the 1st author's time outside of her time at McKinsey. There was no funding, pay or other commercial interest from the participating McKinsey organization. The 4th author, a McKinsey colleague, invited the 1st author's participation in the study. He holds a PhD from King's College in Immunology, is personally interested in this topic, and invests personally and professionally in improving the well-being of colleagues. This study is in no way related to work with clients of the organization. The organization where the research was performed prefers not to be mentioned in the study and hence is anonymized in the paper.

**Competing interests:** The 1st author is a part-time employee of the organization where the research was performed and a part-time affiliated researcher at three Universities. The 4th author is an employee of the same organization and invited the research as part of an employee well-being program. None of the other authors are part of this organization and served as academic researchers. This does not alter our adherence to PLOS ONE policies on sharing data and materials.

A recent study by Accenture reports that 66% of the workforce has experienced mental health challenges, and that about 39% have regular thoughts of suicide [6]. The World Health Organization predicts that mental health-related diseases will be the number-one driver of occupational disability by 2030 [7]. With the accelerating change in the world at large, and the context of work and the content of roles, employees will need to learn and develop at the same pace or faster than before. This increasing need to reskill and to learn new things potentially comes with increasing stress for workers. Getting 'out of the comfort zone' to learn new things may cause stress, and learning the skills as to how to deal with this is an important new requirement for the new world of work [8]. Furthermore, understanding how these skills are related to an overall, more sustainable model for employee well-being will create important insights regarding sustainable career development and lifelong learning [9].

In this study, we focus on the critical soft skill referred to as emotional flexibility, presenting the results from a training intervention study. In so doing, our contribution to extant knowledge is fourfold. First, we explore the new context of knowledge workers. We expect that emotional flexibility training will be particularly relevant to a work setting with private sector knowledge workers. It is known that such work involves high demand on executive brain functioning, and that regulating emotions will be highly beneficial. Hence, tailoring to workplace language and execution of the training to this setting is likely to be essential for engagement and effectiveness. Although progress in workplace mental health and mindfulness training is made (see, e.g., public sessions published at YouTube by Google and LinkedIn), concepts such as emotional flexibility and mindfulness are still approached with a dose of skepticism or even frowned upon in the majority of private sector organizations. Based on prior studies and extant insights, we expect that emotional flexibility can be developed over time through workshop-based training and related self-help tools.

Second, a comprehensive measurement instrument capturing all six sub-processes emotional flexibility is lacking, to date. Acceptant and Commitment Therapy (ACT) argues that six sub-processes of emotional flexibility are key. We test and validate a recently developed instrument for measuring emotional flexibility and the associated sub-processes, and do so in the new environment of knowledge work. We will use Rolffs et al.'s twelve-dimensional Multidimensional Psychological Flexibility Inventory (MPFI) [10]. This instrument consists of six dimensions for emotional flexibility and six for emotional inflexibility, designed to capture both the negative and positive sides of the six ACT sub-processes. In their first article in 2016, Rogge et al. showed very promising outcomes and distinctive factor structure validity with this new instrument in multiple non-worker samples. Since then, we are aware of two additional studies, one regarding the cross-validation with other instruments measuring emotional flexibility [11], and one testing factor structures and validity in Japanese, traditional Mandarin, and simplified Mandarin [12]. This instrument has not been tested before in a workplace environment.

Third, to the best of our knowledge, this is the first time that this new and comprehensive measurement instrument (MPFI) is tested in the context of a longitudinal design. Rogge and colleagues [11] called for future research with a longitudinal design to identify directions of causality. In the current study, this is what we do by adopting an experimental emotional training workshop intervention design with before and after-treatment measures of both emotional flexibility (and inflexibility) and an important outcome variable: General self-efficacy. The latter feeds into our final contribution. That is, fourth, we examine the relation between emotional (in)flexibility and General self-efficacy in the working context. In so doing, we further explore the predictive validity of emotional (in)flexibility and the MPFI.

This paper is organized as follows. After a brief theoretical discussion of this emotional flexibility concept, we move to core of the current paper: an experimental training intervention

study into the effectiveness of emotional flexibility training in a knowledge worker context, and the effect on an important outcome variable: General self-efficacy. After organizing access to a sample of highly-educated knowledge workers, we designed and executed a two-step or cross-over experimental design. In the first step, a treatment group went through an emotional flexibility training intervention, and a control group did not, administering a before-treatment measurement in both groups. In the second step, after after-treatment measurement, the control group received the emotional flexibility training as well, again followed by an after-treatment measurement.

## Emotional flexibility

Emotional flexibility (EF) is a concept central to so-called 'ACT' (Acceptance and Commitment Therapy). ACT is a contextual and applied theory based on Relational Frame Theory (RFT), founded and developed by Steven Hayes and colleagues in the late 1990s [13]. This theory and therapy are part of the third-generation cognitive behavior theories and therapies, and have been successfully applied in clinical psychological therapeutic settings to treat, for example, anxiety, depression, stress, and pain regulation. In 2011, ACT has been officially recognized as evidence-based in the US [14]. The ACT approach focuses on six cognitive sub-processes that are argued to form emotional flexibility: purpose and values, present moment awareness, acceptance, defusion, self-in-context, and committed action. At its core is the focus on helping patients to move toward what is important to them (aligned with their values), instead of moving away (avoidance behavior), as the latter may eventually lead to a reduced quality of life. For example, a patient with social anxiety learns how to still engage with friends and in a job despite anxiety, as opposed to withdraw from social engagement and potentially becoming isolated. This theory centers on a set of sub-processes that help the patient to regulate negative emotions that prevent from fully engaging with life. We briefly elaborate on each of the six sub-processes, and will highlight both the emotional flexibility (EF) and inflexibility (EI) perspectives [10].

### Values

Understanding what is important to you as a human being and what your values are in life and at work, can be very helpful to down-regulate stress when difficult tradeoffs are faced and *ditto* decisions must be made. 'Values' can function as 'anchors' in times of stress and insecurity, and provide a sense of safety. The EF side of this sub-process focuses on being connected with values and what is important, and the ability to prioritize these in times of challenge. EI does refer to not being connected with these values, and losing sight of purpose when things become tough.

### Present moment awareness

The process of present moment awareness, oftentimes called 'Mindfulness', refers to being in the here and now. Having present moment awareness implies being able to notice what is going on. When you have present moment awareness, you are not thinking about tomorrow or yesterday, but accept what is here now. EF relates to 'Mindfulness' (or present moment awareness), which refers to being in the moment, and being aware of and being in touch with emotions and feelings. EI implies being on the automatic pilot, and not paying attention to emotions and feelings.

### Acceptance

Facing difficult thoughts and emotions with nurturing care and self-compassion, as opposed to ignore or avoid them, are core of the 'Acceptance' sub-process. Activities to being aware of,

accept and stay with the emotions, as opposed to avoid them and 'numb' feelings with alternative activities (such overeating, overworking, or alcohol), will help in the processing of these emotions, and to reduce their potentially negative impact. Exposure to difficult emotions and fears, possibly through detachment strategies, will allow a new and manageable 'connection' with the current challenge. EF includes openness to difficult thoughts and emotions, making peace with them rather than suppressing them. EI means resorting to distraction or avoidance as an effort (consciously or unconsciously) to make the negative thoughts and feelings disappear.

### Defusion

Central in the 'Defusion' sub-process is 'detached observation', and disconnecting unhelpful personalized identification with a thought. Understanding that a 'thought' is just a 'thought' and not a truth, individuals learn techniques to distance themselves from these unhelpful thoughts so that these thoughts have less influence or impact. EF refers to being able to experience negative thoughts, but not being caught up in them, allowing such thoughts to be there without being distractive. EI means getting caught up with negative thoughts, identifying with them, and letting them interfere with what is important to you as a person.

### Self-as-context

The sub-process of 'Self-as-context' has to do with seeing life and yourself in context and in the 'grand scheme of things'. One event does not define you, but is part of a tapestry of experiences. Taking a bird's eye view, reframing your experience, and taking a flexible perspective helps to contextualize. EF implies being able to see things in perspective and from a broader point of view. EI, which is also called 'Self-as-content', refers to losing sight of context, and being highly critical to the self for having certain thoughts.

### Committed action

The sub-process of 'Committed action' is about giving yourself the gift of growth and movement toward what matters to you in a conscious way. This sub-process is about leaning into challenging situations where it matters, and setting intentions in a way that helps you to create a sense of 'safety' and self-empowerment. EF involves being focused on what matters, and continuing to work toward these goals. EI refers to giving up on moving towards goals, and hence to inaction and getting derailed from plans.

## Emotional flexibility in the workplace

Because of ACT's pragmatic and contextual perspective, in which a focus on action based on values is central, this approach and associated thinking are highly relevant for the business context setting as well. As far as we are aware, the first ACT study in the working context was published in 2000 [15]. They found improved mental health outcomes and a tendency to innovate after ACT training in a media organization. Since then, quite a few studies have been done within the working context, reporting that that EF is positively related to, e.g., mental health and job performance [16], mental health and physical well-being [17], prejudice reduction [18], learning ability, mental health, and job performance [19], mental health and absence rates [20], stress, work satisfaction, emotional exhaustion, depersonalization, general health, vitality, social functioning, and emotional functioning [21], and reduced emotional exhaustion and strain [22]. However, to the best of our knowledge, no studies have yet been done in the high-paced and highly challenging for-profit knowledge worker context.

In the current study, we investigate the feasibility and effectiveness of training EF amongst knowledge workers, and how such training is associated with General self-efficacy as a potential outcome. Partly, we will replicate prior work executed in the working context, which revealed that EF can be improved by training in a workshop format in a professional setting [23]. A study by Flaxman and Bond [24] found that the ACT training not only improved local government workers' mental health, but also that the improvements were larger for those participants who experienced a higher psychological distress at the start of the program, and appeared to benefit more. Other studies looking into the effectiveness of ACT training in the workplace setting showed similarly positive results for, e.g., social workers [25], health care workers [26], and government workers [22].

The evidence for workplace interventions and effectiveness of ACT training from this little stream of research is highly encouraging [23, 27]. However, many workplace-related studies have focused on healthcare or public sector settings. Studies assessing the effectiveness of EF training in the private sector environment are relatively scarce. Effectiveness was found in the financial sector amongst customer service center workers [16], and in samples from call center employees [19], a corporate call center [20], a media organization [15], and middle managers in medium and large-sized companies [28]. Based on this prior work, we expect similarly positive training effects in our new context of knowledge workers.

*H1*: *Emotional flexibility can be developed through workshop-based training and self-help tools over time.*

Note that the opposite prediction holds for emotional inflexibility, *mutatis mutandis*. As we are not aware of any prior work relating to potentially differential effects of training on the different sub-processes, we refrain from formulating *a priori* hypotheses here. Rather, we let the data speak, and explore this issue inductively.

Above and beyond the development of EF skills, we also expect that learning how to effectively deal with stress in the working context will increase General self-efficacy (GSE). A construct that has been tested in working environments and that is closely related to GSE, is Psychological Capital or PsyCap [29]. PsyCap represents the four sub-constructs of hope, efficacy, resilience, and optimism. Research by Luthans and colleagues found that employee PsyCap is positively related to job satisfaction, organizational commitment, and psychological well-being, and negatively associated with cynicism, turnover intentions, job stress, and anxiety. *"General self-efficacy (GSE) is the belief in one's competence to tackle novel tasks and to cope with adversity in a broad range of stressful or challenging encounters, as opposed to specific self-efficacy, which is constrained to a particular task at hand"* [30] (page 80). GSE is negatively associated with depression, anxiety, and helplessness, and positively related to optimism, self-regulation, and self-esteem in a study across five countries [30].

Our EF workshop training format (see below) provides tools to support the development of a mindset to help dealing with difficult and stressful situations. The overarching notion involves understanding and really connecting with what is important to you, in combination with developing the emotional regulation skills to deal with difficult emotions that arise whilst moving toward what matters. This includes learning something new and moving out of your 'comfort zone'. Prior research found that worksite stress management training amongst government employees was particularly effective for workers with low baseline levels of work-related self-efficacy in combination with a high baseline level of intrinsic motivation [31]. In the context of highly educated knowledge workers, we expect high average intrinsic work motivation. Indeed, a study of Deal et al. [32] reports higher levels of intrinsic motivation amongst higher ranks of professional management versus those lower in the hierarchy, irrespective of generational cohort.

Knowledge workers are expected to have high intrinsic motivation, on average [33]. Hence, we expect that EF and GSE are positively associated. We also expect to see a positive effect of the development of EF skills over time on GSE improvement. Furthermore, although we will not be able to replicate exactly what Lloyd et al. [31] have done, we will explore the difference in increase of both EF and GSE compared to their baselines. Given the profile of the participants in our study, we expect a steeper increase in both EF and GSE for participants with lower baselines. Finally, a recent study of Rogge et al. [11] reveals that emotional flexibility is strongly linked to well-being, whereas emotional inflexibility shows a stronger linkage to psychological distress. GSE is a concept that represents the subjective feeling of being able to deal with difficult and unforeseen situations. It indicates a feeling of resourcefulness to handle whatever happens. This, we expect, will give both an experience of well-being, as well as a reduction in the experience of distress.

***H2a***: *Emotional flexibility is positively related to General self-efficacy.*

***H2b***: *Improvement in emotional flexibility is associated with improvement in General self-efficacy.*

***H2c***: *The improvement of emotional flexibility and General self-efficacy is expected to be steeper for participants with lower baselines.*

Again, as above, the opposite holds for emotional inflexibility. And gain, we will explore potentially different effects for emotional (in)flexibility's sub-processes inductively, given lack of prior work on this.

## Methodology

The project was approved by the internal review board of Maastricht University in the Netherlands. The participants in our study are knowledge workers from a private organization in Germany. We were asked by senior leadership of this organization to investigate the effect of (1) EF training and (2) the impact on parameters of health. To do so, we designed a two-step study. In the first step, we collected data through online surveys before and after an emotional flexibility workshop (non-)treatment. In the second step, which was highly exploratory, we invited a medical doctor to take heart rate variability (HRV) measures. The current paper focuses on the first step only. We recruited the participants with a flyer that was distributed digitally to all eligible colleagues in this organization. Participants could subscribe via e-mail on a first-come-first-serve basis. The limit for workshop participation was set at 30 participants, given capacity constraints on location and to keep the facilitation manageable. Relatively quickly, the workshops were filled, and for both dates people were placed on waiting lists. To mitigate the small sample size and cross-sectional design limitations of much extant work in this area, we opted for a cross-over control group design.

To realize this, we recruited for two groups: one group that started with workshops in February (Group 1) and the second group in April (Group 2), with the latter Group 2 serving as a control group *vis-à-vis* treatment Group 1. Both groups joined a kick-off conference call to explain the process and align mutual expectations. Both groups filled out a pre-process survey at $t1$. Group 1 then went through a process where participants had an individual HRV measurement before starting the first workshop. HRV measurement sessions took about 20 minutes, including the intake and brief post-session conversation in which their results were shared. The actual HRV measurement took 10 minutes. After that, three short worshops followed, which reflect the treatment central to this paper's study. The first and second workshop took about 4.5 hours, including a 30-minutes break. The last booster workshop was about 2.5

hours. Similar to the 2+1 design of Flaxman et al. [23], which is based on Barkham and Shapiro [34], we designed the intervention into three relatively short workshops, providing guidance for simple 'take home' practices and activities in between those sessions. In our design, the first two workshops were about one week apart, and the third workshop followed approximately six weeks later, with take home exercises and one encouragement follow-up email in between all sessions.

We were acutely aware of the utmost importance to follow careful guidelines in research in general (whether this applies to medical data or not). To abide by these guidelines, we took the following steps: (1) Approval: The project was approved by the internal review board of the university; (2) Medical data collection (not used in the current paper): we made sure to invite a medically schooled professional (MD qualified) to collect this data, following the strict rules of patient data collection that are normal in this profession; (3) Participant Consent: The consent issue was emphasized in each step of the study. That is, participants were notified on different occasions about the research elements of the study, and the option to exit, as each and every step in the process is voluntary: (a) at the kick-off call explaining expectations and signing up; (b) when completing the survey; (c) during the workshops; and (d) during the HRV measurement when the medical doctor recorded their verbal consent (not used in this study). Because of the different stages in the study we chose to note verbal consent instead. For the overall project we received written consent of the organization.

Although certain elements overlapped with the protocol of Flaxman et al. [23], we based the content design of the workshops on extensive practical experience with and piloting of the workshop format introduced in Brassey et al. [8], with a strong focus on aligning content with the context of knowledge workers. Furthermore, as said, we used the detailed and validated twelve-dimensional psychological (in)flexibility scale from Rolffs et al. [10]. We learned through pilot workshops that explaining to the audience all of the six sub-processes of flexibility and inflexibility is key, in combination with translating ACT insights to resonate with the practice of the participants' professional knowledge context. Indeed, with our German sample, the combination with the questionnaire of Rolffs et al. [10] and a closer alignment to the working context of these knowledge workers worked very well. The training sessions were performed by the first author, who is an experienced executive learning and development practitioner, and an academic researcher in EF, and who is officially trained as an ACT facilitator.

After both the first and second workshop, participants received a short email with relevant follow-up information. These short emails, which we called 'whisper courses', were gentle nudges to remind the participants to work on their progress through practicing with the take home instructions. During this period, the second group was not going through workshops, but was exposed to an informal campaign around better performance at work. We explicitly asked all group members not to communicate about the content of the workshops with their colleagues. Given our request and the nature of the work they do, implying that they are traveling during the week away from their office, we think there is a very low chance that the control Group 2 knew of the content before going through the workshops themselves. Both groups filled out a post-workshop survey. After that, the second Group 2 went through the same process. S1 Fig visualizes the whole process.

## Sample and measures

### Sample

As mentioned above, we organized our emotional flexibility workshop series on request of senior leadership of the participating organization. Participants were recruited on a voluntary

basis within an office context. Through up-front written information, explicit confirmation during the kick-off calls, and the introduction of the survey, all participants were informed that all data and insights collected from during the process would be used for internal purposes and scientific research only. Individual data would be kept confidential to the researchers, and HRV data would only be collected and processed by a qualified medical doctor. We explicitly announced that voluntary participation in the program would imply giving consent to this approach.

Both groups were oversubscribed at the start. For both groups, participants on the reserve list were invited to fill out the surveys as well, but only those that participated in at least one of the three workshops were included in the before and after-workshop tests. Along the way, substantial attrition across the workshops emerged, which is unfortunately the reality of the working context (with serving clients being priority number one). That is, the participants often had external obligations at client sites that they had to attend to last minute. Therefore, during each of the workshops, we always kicked off with a brief recap what had already been done, and relevant reading materials with summaries of the involved workshop were shared with all participants. Below, we visualize the participation in terms of size and attrition per step in the process. (see S2 Fig).

### Emotional flexibility

To measure emotional flexibility, we used the multidimensional psychological flexibility inventory (MPFI) developed by Rolffs et al. [10]. This inventory has twelve dimensions: six for EF (= emotional flexibility) and six for EI (= emotional inflexibility). To date, this is the only available inventory that measures all six sub-processes of emotional flexibility, being associated with good reliability and validity. The questionnaire consists of 60 items in total, all rated, in line with recommendations of the developers, on a six-point scale from 'Never true' (1) to 'Always true' (6). (see S1 Appendix). We administered the survey at the start of the overall process to all participants within both groups. Subsequently, after the first series of workshops that were only attended by the first Group 1, both groups filled out a survey again. A third survey was sent to the second Group 2 after they went through their series of workshops. In that way, we were able to use the survey results of the second Group 2 as a quasi-control *vis-à-vis* the first Group 1's treatment for emotional flexibility progress (and / or the reduction of emotional inflexibility).

### General self-efficacy

The well-established General self-efficacy (GSE) scale [35] was used to measure this important outcome variable. This scale includes ten items. (see S1 Appendix). Example items are: 'I can always manage to solve difficult problems if I try hard enough' and 'I can usually handle whatever comes my way'. All ten items are rated on a four-point scale, in line with the recommendations by the developers: 1 = 'Not at all true'; 2 = 'Hardly true'; 3 = 'Moderately true'; and 4 = 'Exactly true'.

## Results

### Overview

The descriptive statistics of the survey variables of the first questionnaire for the first and second group together are presented in Table 1, specifically means, standard deviations and Spearman's rho correlations. We include our central variables regarding emotional (in)flexibility and General self-efficacy, of which we introduce the psychometrics below.

The Variance Inflation Factor (VIF) ranged from 1.129 to a maximum of 4.593, which is far below the critical threshold of 10. Hence, multicollinearity is not an issue. Of our 59 participants, 44% is female, with the majority of participants (71%) in the age range of 25–34 years and 25% within the 35–44 year range. One participant was in the age range of 18–24 years and one in the range of 45–64 years. Most are highly educated, with one participant indicating to have some college but no degree, 61% having a university degree, and 37% an advanced professional degree (JD, MD, et cetera) or doctorate (PhD).

## Psychometric analysis

Our psychometrics are conducted with the data from the first survey responses from both the first and second group together, as administered before the training. Strictly speaking, the total *n*-size for this survey is 59 is too low for a proper factor analysis with 12 dimensions and 60 items [36]. For exploratory purposes, and with reference to earlier evidence provided by Rogge et al. [11], we performed a principal components analysis on the twelve dimensions of the multidimensional psychological flexibility inventory anyway. To our surprise, the initial principal components analysis showed quite a good factorial structure for such a small *n*-size, indicating 11 dimensions, as reported in the Appendix (see S2 Appendix). When we re-ran the analysis to confirm 12 dimensions, the pattern analysis continued to lump two dimensions of EF together: 'Values', and 'Committed action'. All other dimensions loaded on their respective factor (see S3 Appendix). We considered this a good representation of the factorial structure to continue with our next steps, with all Cronbach Alpha values being 0.9 or above.

**Wilcoxon signed rank tests EF and EI.** We performed Wilcoxon signed rank tests on the overall scales of EF, EI and the respective sub-processes. To do this, we selected only those participants who joined at least one workshop. The results for the first Group 1 are presented in Table 2. If the emotional flexibility training is effective, we expect to see increases in the EF scores and decreases in the EI scores. Indeed, the data reveal this expected pattern in the tests (*n* = 21). Actually, we noticed that the means for all sub-processes increase for EF and decrease for EI, with six of these changes being significant: One EF sub-process ('Defusion'), EI overall and four EI sub-processes ('Avoidance', 'Self-as-content', 'Fusion', and 'Inaction'). For this first group, the workshops seemed to be mostly effective to decrease EI. Also, the aggregated scores for EF and EI changed in the expected direction, but only the EI change is significant.

Subsequently, we compare these results with the control group's (Group 2), which had not gone through the workshops, but which experienced an informal campaign regarding better performance from their organization (not in group or workshop format). The results are presented in Table 3. The total control Group 2 includes 27 participants. The before and after mean for three dimensions ('Self as context', 'Self as content', and 'Fusion') deteriorated for this group. Overall EI and sub-process 'Experiential Avoidance' approach significance. We only find two significant results: an increase in the EF sub-process of 'Values' and in the EI Sub-process of 'Lack of contact with Values' We cannot explain why these two scores improved. Perhaps, this was an impact of the campaign that was happening in the organization, but then we would also expect to see this improvement for Group 1. Taking a closer look at the baseline results, we see that treatment Group 1 had already a much higher baseline of overall emotional flexibility compared to the control Group 2; although also Group 1 is associated with an increased mean EF score, this increase is not significant. We will return to this baseline issue later.

To explore this lower effect for the control Group 2 compared to treatment Group 1 further, we also analyzed the before and after-workshop results of Group 2, going through the series workshops in the next phase. These results are presented in Table 4.

**Table 1. Means, standard deviations and bi-variate spearman rho correlations.**

|      | M     | SD    | GSE     | EF      | A       | M       | SACX    | D       | V       | CA      | EI      | EA     | LM      | SACN    | F       | LV    | I   |
|------|-------|-------|---------|---------|---------|---------|---------|---------|---------|---------|---------|--------|---------|---------|---------|-------|-----|
| GSE  | 3.186 | 0.413 | 1       |         |         |         |         |         |         |         |         |        |         |         |         |       |     |
| EF   | 3.600 | 0.834 | .435**  | 1       |         |         |         |         |         |         |         |        |         |         |         |       |     |
| A    | 3.231 | 1.016 | 0.064   | .650**  | 1       |         |         |         |         |         |         |        |         |         |         |       |     |
| M    | 3.681 | 1.127 | 0.223 † | .757**  | .683**  | 1       |         |         |         |         |         |        |         |         |         |       |     |
| SACX | 4.000 | 0.953 | .423**  | .802**  | .341**  | .455**  | 1       |         |         |         |         |        |         |         |         |       |     |
| D    | 3.237 | 1.059 | .277*   | .666**  | .472**  | .452**  | .578**  | 1       |         |         |         |        |         |         |         |       |     |
| V    | 3.600 | 1.112 | .386**  | .826**  | .396**  | .503**  | .671**  | .379**  | 1       |         |         |        |         |         |         |       |     |
| CA   | 3.851 | 0.991 | .534**  | .826**  | .373**  | .517**  | .670**  | .407**  | .827**  | 1       |         |        |         |         |         |       |     |
| EI   | 2.881 | 0.689 | -.314*  | -.597** | -.275*  | -.323*  | -.449** | -.516** | -.560** | -.591** | 1       |        |         |         |         |       |     |
| EA   | 3.373 | 0.974 | 0.076   | -0.03   | -0.13   | -0.1    | 0.158   | 0.089   | -0.06   | -0.06   | .293*   | 1      |         |         |         |       |     |
| LM   | 2.864 | 1.194 | -.369** | -.587** | -.337** | -.561** | -.302*  | -0.18   | -.541** | -.635** | .623**  | .288*  | 1       |         |         |       |     |
| SACN | 2.359 | 1.088 | -0.110  | -0.17   | -0.07   | -0.02   | -0.15   | -.316*  | -0.06   | -0.19   | .689**  | 0.200  | 0.243 † | 1       |         |       |     |
| F    | 2.773 | 1.159 | -0.17   | -0.256 †| 0.078   | 0.105   | -.362** | -.514** | -0.23   | -.313*  | .684**  | -0.01  | 0.139   | .575**  | 1       |       |     |
| LV   | 3.403 | 1.063 | -0.16   | -.308*  | -0.2    | -0.12   | -0.16   | -0.1    | -.421** | -.352** | .568**  | 0.07   | .362**  | 0.216   | 0.218   | 1     |     |
| I    | 2.512 | 0.984 | -.474** | -.433** | -0.15   | -0.18   | -.500** | -.530** | -.381** | -.473** | .573**  | -0.15  | 0.183   | .365**  | .509**  | 0.205 | 1   |

n = 59

** Correlation (two-tailed) is significant at the 0.01 level

* Correlation is significant at the 0.05 level

† Correlation is significant at 0.1 level; Variables: A = Acceptance; M = Present moment awareness; SACX = Self-as-context; D = Defusion; V = Purpose and values; CA = Committed action; EA = Experiential avoidance; LM = Lack of present moment awareness; SACN = Self-as-content; F = Fusion; LV = Lack of contact with values; I = Inaction; EF = Emotional flexibility; EI = Emotional inflexibility; and GSE = General self-efficacy.

We have an *n*-size of 17 participants who both filled out the before and after surveys, and who joined for at least one workshop. We reveal significant increases or respective decreases

**Table 2. Wilcoxon signed rank tests for EF, EI and sub-processes for treatment Group 1.**

|           | Negative ranks | | | Positive ranks | | | Test statistics | | |
|-----------|------|-----------|--------------|------|-----------|--------------|------|--------|-------|
|           | n    | Mean rank | Sum of ranks | n    | Mean rank | Sum of ranks | Ties | Z      | p     |
| **EF2-EF1** | 8  | 9.50      | 76           | 13   | 11.92     | 255          | 0    | -1.373 | 0.170 |
| A2-A1     | 7    | 7.36      | 51.5         | 11   | 10.86     | 119.5        | 3    | -1.485 | 0.138 |
| M2-M1     | 7    | 10.21     | 71.5         | 14   | 11.39     | 159.5        | 0    | -1.535 | 0.125 |
| SACX2-SACX1 | 8  | 9.75      | 78           | 10   | 9.3       | 93           | 3    | -0.327 | 0.743 |
| D2-D1     | 6    | 11        | 66           | 15   | 11        | 165          | 0    | -1.724 | **0.085** |
| V2-V1     | 8    | 10.06     | 80.5         | 11   | 9.95      | 109.5        | 2    | -0.585 | 0.559 |
| CA2-CA1   | 10   | 11.65     | 116.5        | 11   | 10.41     | 114.5        | 0    | -0.035 | 0.972 |
| **EI2-EI1** | 16 | 12.5      | 200          | 5    | 6.2       | 31           | 0    | -2.941 | **0.003** |
| EA2-EA1   | 13   | 9.73      | 126.5        | 5    | 8.9       | 44.5         | 3    | -1.791 | **0.073** |
| LM2-LM1   | 11   | 9.91      | 109          | 7    | 8.86      | 62           | 3    | -1.032 | 0.302 |
| SACN2-SACN1 | 13 | 10.04     | 130.5        | 5    | 8.1       | 40.4         | 3    | -1.968 | **0.049** |
| F2-F1     | 12   | 11.46     | 137.5        | 7    | 7.5       | 52.5         | 2    | -1.717 | **0.086** |
| LV2-LV1   | 11   | 13        | 143          | 9    | 7.44      | 67           | 1    | -1.421 | 0.155 |
| I2-I1     | 15   | 9         | 135          | 2    | 8         | 18           | 4    | -2.783 | **0.005** |

n = 21; Variables: EF = Emotional flexibility; A = Acceptance; M = Present moment awareness; SACX = Self-as-context; D = Defusion; V = Purpose and values; CA = Committed action; EI = Emotional inflexibility; EA = Experiential avoidance; LM = Lack of present moment awareness; SACN = Self-as-content; F = Fusion; LV = Lack of contact with values; and I = Inaction.

**Table 3. Wilcoxon signed rank tests for EF, EI and subscales control Group 2.**

|  | Negative ranks | | | Positive ranks | | | Test statistics | | |
|---|---|---|---|---|---|---|---|---|---|
|  | n | Mean rank | Sum of ranks | n | Mean rank | Sum of ranks | Ties | Z | p |
| **EF2-EF1** | 12 | 13.04 | 156.5 | 15 | 14.77 | 221.5 | 0 | -0.781 | 0.435 |
| A2-A1 | 11 | 11.23 | 123.5 | 13 | 13.58 | 176.5 | 3 | -0.761 | 0.447 |
| M2-M1 | 12 | 12.04 | 144.5 | 14 | 14.75 | 206.5 | 1 | -0.790 | 0.429 |
| SACX2-SACX1 | 14 | 0.82 | 137.5 | 8 | 14.44 | 115.5 | 5 | -0.360 | 0.719 |
| D2-D1 | 10 | 9.35 | 93.5 | 11 | 12.50 | 137.5 | 6 | -0.770 | 0.441 |
| V2-V1 | 7 | 12.21 | 85.5 | 17 | 12.62 | 312.5 | 3 | -1.858 | **0.063** |
| CA2-CA1 | 12 | 11.71 | 140.5 | 14 | 15.04 | 210.5 | 1 | -0.892 | 0.372 |
| **EI2-EI1** | 17 | 15.31 | 258.5 | 10 | 11.95 | 119.5 | 0 | -1.670 | 0.095 |
| EA2-EA1 | 15 | 13.87 | 208 | 9 | 10.22 | 92 | 3 | -1.662 | 0.097 |
| LM2-LM1 | 14 | 17.39 | 243.5 | 13 | 10.35 | 134.5 | 0 | -1.314 | 0.189 |
| SACN2-SACN1 | 10 | 9.7 | 97 | 12 | 13 | 156 | 5 | -0.961 | 0.336 |
| F2-F1 | 9 | 11.83 | 106.5 | 14 | 12.11 | 169.5 | 4 | -0.964 | 0.335 |
| LV2-LV1 | 21 | 14.29 | 300 | 5 | 10.2 | 51 | 1 | -3.172 | **0.002** |
| I2-I1 | 9 | 11 | 99 | 11 | 10.09 | 111 | 7 | -0.226 | 0.821 |

n = 27; Variables: EF = Emotional flexibility; A = Acceptance; M = Present moment awareness; SACX = Self-as-context; D = Defusion; V = Purpose and values;

CA = Committed action; EI = Emotional inflexibility; EA = Experiential avoidance; LM = Lack of present moment awareness; SACN = Self-as-content; F = Fusion;

LV = Lack of contact with values; and I = Inaction.

for nine sub-processes. The scores for all dimensions of EF increased, as well as of three of EI' sub-processes ('Lack of present moment awareness, 'Self-as-content', and 'Fusion'). The changes in both the aggregated scores for EF and EI are significant.

We deductively hypothesized that EF can be developed through workshops and self-help tools over time, and inductively speculated that this may be true for all sub-processes

**Table 4. Wilcoxon signed rank tests for EF, EI and sub-processes Group 2.**

|  | Negative ranks | | | Positive ranks | | | Test statistics | | |
|---|---|---|---|---|---|---|---|---|---|
|  | n | Mean rank | Sum of ranks | n | Mean rank | Sum of ranks | Ties | Z | p |
| **EF3-EF2** | 2 | 5.00 | 10 | 15 | 9.53 | 143 | 0 | -3.148 | **0.002** |
| A3-A2 | 4 | 3.5 | 14 | 11 | 9.64 | 106 | 2 | -2.623 | **0.009** |
| M3-M2 | 0 | 0 | 0 | 15 | 8 | 120 | 2 | -3.417 | **0.001** |
| SACX3-SACX2 | 5 | 5.6 | 28 | 12 | 10.42 | 125 | 0 | -2.303 | **0.021** |
| D3-D2 | 4 | 4 | 16 | 12 | 10 | 120 | 1 | -2.7 | **0.007** |
| V3-V2 | 4 | 4.13 | 16.5 | 9 | 8.28 | 74.5 | 4 | -2.033 | **0.042** |
| CA3-CA2 | 6 | 5.58 | 33.5 | 10 | 10.25 | 102.5 | 1 | -1.797 | **0.072** |
| **EI3-EI2** | 12 | 10.17 | 122 | 5 | 6.2 | 31 | 0 | -2.154 | **0.031** |
| EA3-EA2 | 8 | 9 | 72 | 9 | 9 | 81 | 0 | -0.214 | 0.831 |
| LM3-LM2 | 11 | 7.27 | 80 | 2 | 5.5 | 11 | 4 | -2.415 | **0.016** |
| SACN3-SACN2 | 13 | 9 | 117 | 4 | 9 | 36 | 0 | -1.928 | **0.054** |
| F3-F2 | 12 | 8.46 | 101.5 | 4 | 8.63 | 34.5 | 1 | -1.738 | **0.082** |
| LV3-LV2 | 9 | 9.17 | 82.5 | 6 | 6.25 | 37.5 | 2 | -1.284 | 0.199 |
| I3-I2 | 10 | 7.75 | 77.5 | 5 | 8.5 | 42.5 | 2 | -0.997 | 0.319 |

n = 17; Variables: EF = Emotional flexibility; A = Acceptance; M = Present moment awareness; SACX = Self-as-context; D = Defusion; V = Purpose and values;

CA = Committed action; EI = Emotional inflexibility; EA = Experiential avoidance; LM = Lack of present moment awareness; SACN = Self-as-content; F = Fusion;

LV = Lack of contact with values; and I = Inaction.

**Table 5. Wilcoxon signed rank tests for GSE Group 1, control Group 2 and treatment Group 2.**

| | | Negative ranks | | | Positive ranks | | | Test statistics | | |
|---|---|---|---|---|---|---|---|---|---|---|
| | | *n* | Mean rank | Sum of ranks | *n* | Mean rank | Sum of ranks | Ties | Z | *p* |
| GSE2-GSE1 | Group 1 | 6 | 5.00 | 30 | 11 | 11.8 | 123 | 4 | -2.220 | **0.026** |
| GSE2-GSE1 | Control Group | 11 | 11.95 | 144.5 | 12 | 12.04 | 144.5 | 4 | -0.200 | 0.842 |
| GSE3-GSE2 | Group 2 | 6 | 4.42 | 26.5 | 11 | 11.5 | 126.5 | 0 | -2.386 | **0.017** |

separately, too. The results show that treatment Group 1 is associated with significant progress on one of the sub-processes of EF, four of the six sub-processes of EI, as well as on the aggregated dimension of EI. During the same period, the results for the control Group 2 were inconsistent (and only improved significantly for 'Values' and 'Lack of contact with values'). The control Group 2 reveals much more progress after they went through the series of workshops–namely on nine of EF and EI's sub-processes, and on both aggregated scores for EF and EI. The control Group 2 improved mostly on EF and on half of the EI scores, whereas progress of treatment Group 1 runs primarily through a reduction in EI. Also notable is the direction of the improvements: these are only after treatment (i.e. after the training interventions) consistently in the direction as expected (either positive for emotional flexibility or negative for emotional inflexibility), which is not the case for the control group results. By and large, these results are in support of Hypothesis H1.

**Wilcoxon signed rank tests for general self-efficacy.** The results of all Wilcoxon Signed Rank tests for General self-efficacy are presented in Table 5.

Group 1 reveals significant improvement of GSE after the workshop interventions compared to before the treatment. The control group (Group 2) is associated with a non-significant *decrease*. After their workshop interventions, Group 2's increase of GSE turns significant, too. This confirms our Hypothesis H2a.

Seeing these overall before and after results for both the treatment groups and the control group, we subsequently looked into any possible effect due to attrition. Regarding the participants who had signed up for Group 1, we eventually had 28 completing the before-intervention survey1. We had an attrition of six participants (21%) who did not come to the first workshop (*n* = 22 for survey1 and workshop1 participants) and lost another five (18%) for the after-intervention survey2 (total *n* = 17 for both surveys and at least one workshop, total attrition vis-à-vis first survey: 39%). The six who did not show up but who did fill out the survey scored lower on mean EF (3.02) than the 'remainers' (3.84), higher on mean IF (3.25) compared to the 'remainers' (2.69), and slightly lower on mean GSE (2.95) compared to 'remainers' (3.18)

We can reasonably argue that the six who could not come would actually need the workshop more, on average, than the employees who came, given their slightly less positive results. We did get cancellations via e-mail for some of those six, and know this had to do with personal circumstances or work challenges. One explanation we can think of is that having lower flexibility, higher inflexibility and lower GSE can perhaps lead to less courage to push back when work challenges arise, and not daring to withstand leadership. But we really have only six datapoints for this without any further information. Hence, all this is highly speculative. In terms of demographics, the six who did not show up consisted of three females and three males. One of these participants was from the age group 18–24, four of these participants were from the age group 25–34 years, and one from 34–44 years. There were five participants with the German nationality and one with the Austrian nationality. Finally, one participant indicated to have a Bachelor's degree, three a Master's degree and two had a PhD degree. This is not substantially different from their colleagues who did show up.

Regarding Group 2, from the participants who had completed survey1 ($n$ = 31), we eventually had 27 filling in the second survey (representing the control group). Subsequently, 23 (15% attrition vis-à-vis the second survey) of those participants came to the first workshop, and we lost another two for the after-intervention survey2 (total $n$ = 21 for both surveys and at least one workshop; total attrition of 22% vis-à-vis second survey). We also looked at the averages for these eight participants who filled out the survey at the start, but who eventually did not join the first workshop, compared to the rest of the group that joined. The comparisons were as follows: EF ('joiners': 3.32; attrition: 4.16), EI ('joiners': 2.95; attrition: 2.95), and GSE ('joiners': 3.25; attrition: 3.23).

The 'joiners' started with a lower EF than their colleagues who did not come, eventually, but the EI and GSE scores were the same. This may indicate that they realized, perhaps, that they were not really interested after all. As for Group 1, from some we got cancellations indicating they had to cancel, with reasons as above. The attrition group demographics were as follows: three of these were females, and five males; six were in the age group 25–34 years, and two in the age group 34–44. All were of German nationality. Seven of them had a Master's degree and one a PhD degree. Again, the differences with the 'joiners' were not large.

As an aside, we would like to argue that the above does not imply huge attrition, given our field setting of a private organization. The attrition was not so much related to survey response itself; these were actually not that bad at all. For Group 1, total attrition was 39% (vis-à-vis those who completed the first survey and joined at least one workshop); for Group 2, this was 22%. The biggest attrition was seen regarding joining workshops, and not so much for completing surveys. Four of the eight who did not join workshops still filled out the last survey, an indication that they were committed to actually participate. Key here is that surveys could be done everywhere anytime, since they were online. Given their very busy jobs and the agreement that the job came first, we should not be surprised that quite a few could not physically join the workshop after all.

To answer the next Hypothesis H2b that improvement in EF is associated with improvement in GSE, we calculated the delta-differences of the results for all respective overall scores and their sub-processes by subtracting the second result from the first (for example, EFt2 – EFt1 = ΔEF). Subsequently, we looked at the bivariate correlations between those delta-differences. The results for Group 1 ($n$ = 21) are presented in Table 6, and for Group 2 ($n$ = 17) in Table 7.

For Group 1, we see that the improvement of GSE results are positively associated with improvement results of overall EF and four sub-processes: 'Acceptance', 'Present moment awareness', 'Values', and 'Committed action'. Also, we observe negative correlation with 'Lack of present moment awareness'. Overall, for Group 1, the strongest association is that between the deltas of GSE and EF.

For Group 2, we reveal a positively significant correlation for the sub-processes of 'Self-as-context' and 'Committed action'. The correlations between the delta of GSE EF and EI are nearing significance. The relationship with GSE is positive and significant for 'Experiential Avoidance', and negative and significant for 'Fusion', 'Lack of contact with values', and 'Inaction'. Overall, we observe a stronger effect for the reduction of emotional inflexibility in association with the increase of GSE for Group 2. In all, most correlations are as expected, with the exception of the association between GSE and the EI sub-process 'Experiential Avoidance', which is significant but in the opposite direction than expected. Future research is needed to see whether or not this counterintuitive finding will survive replication. In all, these results provide partial support for Hypothesis H2b: Improvement of GSE is associated with improvement of EF and EI. The interesting differences in patterns across groups are directed in our test of Hypothesis H2c regarding the potential impact of baseline differences.

**Table 6. Delta correlations Group 1 for improvement EF, EI and GSE.**

| Var. | Corr. | EF | EI | GSE | A | M | SACX | D | V | CA | EA | LM | SACN | F | LV | IA |
|------|-------|------|------|------|------|------|------|------|------|------|------|------|------|------|------|------|
| EF | r | 1.000 | | | | | | | | | | | | | | |
| | Sig. | | | | | | | | | | | | | | | |
| EI | r | -0.359 | 1.000 | | | | | | | | | | | | | |
| | Sig. | 0.110 | | | | | | | | | | | | | | |
| GSE | r | .548* | -0.242 | 1.000 | | | | | | | | | | | | |
| | Sig. | 0.010 | 0.290 | | | | | | | | | | | | | |
| A | r | .803** | -0.299 | .456* | 1.000 | | | | | | | | | | | |
| | Sig. | 0.000 | 0.188 | 0.038 | | | | | | | | | | | | |
| M | r | .752** | -0.032 | .468* | .674** | 1.000 | | | | | | | | | | |
| | Sig. | 0.000 | 0.891 | 0.033 | 0.001 | | | | | | | | | | | |
| SACX | r | .748** | -0.209 | 0.288 | 0.411 | .458* | 1.000 | | | | | | | | | |
| | Sig. | 0.000 | 0.363 | 0.205 | 0.064 | 0.037 | | | | | | | | | | |
| D | r | .816** | -0.274 | 0.182 | .754** | .622** | .553** | 1.000 | | | | | | | | |
| | Sig. | 0.000 | 0.229 | 0.429 | 0.000 | 0.003 | 0.009 | | | | | | | | | |
| V | r | .731** | -0.388 | .506* | 0.369 | .457* | .472* | 0.398 | 1.000 | | | | | | | |
| | Sig. | 0.000 | 0.082 | 0.019 | 0.100 | 0.037 | 0.031 | 0.074 | | | | | | | | |
| CA | r | .711** | -0.395 | .610** | .446* | 0.258 | .566** | 0.382 | .596** | 1.000 | | | | | | |
| | Sig. | 0.000 | 0.077 | 0.003 | 0.043 | 0.259 | 0.007 | 0.088 | 0.004 | | | | | | | |
| EA | r | -0.017 | 0.371 | -0.271 | -0.116 | -0.109 | 0.371 | 0.193 | -0.300 | -0.074 | 1.000 | | | | | |
| | Sig. | 0.942 | 0.097 | 0.236 | 0.618 | 0.637 | 0.097 | 0.401 | 0.187 | 0.751 | | | | | | |
| LM | r | -.467* | .640** | -0.400 | -.515* | -0.398 | -0.172 | -0.199 | -.487* | -0.338 | .514* | 1.000 | | | | |
| | Sig. | 0.033 | 0.002 | 0.073 | 0.017 | 0.074 | 0.457 | 0.388 | 0.025 | 0.135 | 0.017 | | | | | |
| SACN | r | -0.273 | .683** | -0.036 | -0.108 | -0.007 | -0.238 | -0.316 | -0.376 | -0.160 | 0.079 | 0.349 | 1.000 | | | |
| | Sig. | 0.230 | 0.001 | 0.876 | 0.641 | 0.977 | 0.299 | 0.163 | 0.093 | 0.487 | 0.733 | 0.121 | | | | |
| F | r | -0.213 | .700** | -0.105 | 0.101 | 0.213 | -0.429 | -0.152 | -0.306 | -0.383 | -0.070 | 0.100 | .493* | 1.000 | | |
| | Sig. | 0.355 | 0.000 | 0.651 | 0.662 | 0.353 | 0.052 | 0.511 | 0.177 | 0.086 | 0.762 | 0.665 | 0.023 | | | |
| LV | r | -0.009 | .637** | 0.158 | -0.173 | 0.117 | -0.013 | -0.188 | 0.233 | -0.013 | -0.071 | 0.162 | 0.407 | 0.429 | 1.000 | |
| | Sig. | 0.969 | 0.002 | 0.493 | 0.454 | 0.615 | 0.957 | 0.415 | 0.309 | 0.956 | 0.758 | 0.483 | 0.067 | 0.053 | | |
| I | r | -0.332 | .541* | -0.248 | -0.260 | 0.070 | -0.295 | -0.328 | -0.192 | -.487* | -0.126 | 0.164 | 0.088 | .577** | 0.295 | 1.000 |
| | Sig. | 0.142 | 0.011 | 0.279 | 0.254 | 0.762 | 0.194 | 0.147 | 0.403 | 0.025 | 0.585 | 0.478 | 0.703 | 0.006 | 0.195 | |

n = 21

** Correlation is significant at the 0.01 level (two-tailed)

* Correlation is significant at the 0.05 level (two-tailed). Variables: EF = Emotional flexibility; EI = Emotional inflexibility; GSE = General self-efficacy; A = Acceptance; M = Present moment awareness; SACX = Self-as-context; D = Defusion; EA = Experiential avoidance; V = Purpose and values; CA = Committed action; LM = Lack of present moment awareness; SACN = Self-as-content; F = Fusion; LV = Lack of contact with values; and I = Inaction.

To answer this next Hypothesis H2c, we must examine whether the improvement of EF, EI, their respective sub-processes, and GSE is higher for participants with a lower baseline. Because of the small sample sizes that we work with, we decided to only conduct exploratory analyses by simply categorizing the data into six categories of the baseline scores, and then look at the average delta per category of participants who started at this respective baseline. The results of this exploratory analysis for Groups 1 and 2 for their aggregated results for GSE, EF and EI before and after their respective workshop interventions are summarized in Table 8.

Again, we emphasize that this analysis is highly exploratory and should be done on larger samples sizes in future research. Looking at the small sample that we have, we indeed see the same patterns for both groups. For Group 1, we observe that if the EF scores start at a lower

**Table 7. Delta correlations Group 2 for improvement EF, EI and GSE.**

| Var. | Corr. | EF | EI | GSE | A | M | SACX | D | V | CA | EA | LM | SACN | F | LV | IA |
|------|-------|----|----|-----|---|---|------|---|---|----|----|----|------|---|----|----|
| EF | r | 1.000 | | | | | | | | | | | | | | |
| | Sig. | | | | | | | | | | | | | | | |
| EI | r | -.842** | 1.000 | | | | | | | | | | | | | |
| | Sig. | 0.000 | | | | | | | | | | | | | | |
| GSE | r | 0.415 | -0.409 | 1.000 | | | | | | | | | | | | |
| | Sig. | 0.097 | 0.103 | | | | | | | | | | | | | |
| A | r | .800** | -.758** | 0.128 | 1.000 | | | | | | | | | | | |
| | Sig. | 0.000 | 0.000 | 0.623 | | | | | | | | | | | | |
| M | r | .846** | -.767** | 0.381 | .712** | 1.000 | | | | | | | | | | |
| | Sig. | 0.000 | 0.000 | 0.132 | 0.001 | | | | | | | | | | | |
| SACX | r | .708** | -.503* | 0.465 | 0.355 | .542* | 1.000 | | | | | | | | | |
| | Sig. | 0.001 | 0.040 | 0.060 | 0.162 | 0.025 | | | | | | | | | | |
| D | r | .815** | -.665** | 0.316 | .507* | .611** | .538* | 1.000 | | | | | | | | |
| | Sig. | 0.000 | 0.004 | 0.217 | 0.038 | 0.009 | 0.026 | | | | | | | | | |
| V | r | .760** | -.627** | 0.129 | .549* | .664** | 0.283 | .750** | 1.000 | | | | | | | |
| | Sig. | 0.000 | 0.007 | 0.621 | 0.023 | 0.004 | 0.271 | 0.001 | | | | | | | | |
| CA | r | .706** | -.573* | .493* | .525* | 0.395 | .519* | 0.425 | 0.382 | 1.000 | | | | | | |
| | Sig. | 0.002 | 0.016 | 0.044 | 0.031 | 0.117 | 0.033 | 0.089 | 0.130 | | | | | | | |
| EA | r | -0.372 | 0.437 | 0.440 | -.581* | -0.280 | -0.105 | -0.165 | -0.445 | -0.138 | 1.000 | | | | | |
| | Sig. | 0.142 | 0.080 | 0.077 | 0.014 | 0.277 | 0.687 | 0.528 | 0.074 | 0.597 | | | | | | |
| LM | r | -.719** | .862** | -0.246 | -.692** | -.516* | -0.399 | -.713** | -0.470 | -.520* | 0.348 | 1.000 | | | | |
| | Sig. | 0.001 | 0.000 | 0.342 | 0.002 | 0.034 | 0.113 | 0.001 | 0.057 | 0.032 | 0.171 | | | | | |
| SACN | r | -.597* | .769** | -0.313 | -.508* | -.696** | -0.374 | -.507* | -.558* | -0.144 | 0.258 | .588* | 1.000 | | | |
| | Sig. | 0.011 | 0.000 | 0.221 | 0.037 | 0.002 | 0.139 | 0.038 | 0.020 | 0.582 | 0.317 | 0.013 | | | | |
| F | r | -.553* | .592* | -.772** | -0.227 | -.593* | -.511* | -.548* | -0.327 | -0.367 | -0.388 | 0.474 | .591* | 1.000 | | |
| | Sig. | 0.021 | 0.012 | 0.000 | 0.380 | 0.012 | 0.036 | 0.023 | 0.201 | 0.147 | 0.124 | 0.054 | 0.012 | | | |
| LV | r | -.514* | .607** | -.549* | -0.360 | -.513* | -0.206 | -0.339 | -0.382 | -.600* | 0.117 | 0.359 | 0.219 | 0.381 | 1.000 | |
| | Sig. | 0.035 | 0.010 | 0.022 | 0.156 | 0.035 | 0.428 | 0.184 | 0.130 | 0.011 | 0.656 | 0.157 | 0.398 | 0.132 | | |
| I | r | -.702** | .840** | -0.436 | -.677** | -.573* | -.503* | -0.458 | -0.329 | -.652** | 0.232 | .742** | .488* | .541* | .520* | 1.000 |
| | Sig. | 0.002 | 0.000 | 0.080 | 0.003 | 0.016 | 0.039 | 0.064 | 0.197 | 0.005 | 0.369 | 0.001 | 0.047 | 0.025 | 0.032 | |

n = 17 ** Correlation is significant at the 0.01 level (two-tailed)

* Correlation is significant at the 0.05 level (two-tailed). Variables: EF = Emotional flexibility; EI = Emotional inflexibility; GSE = General self-efficacy; A = Acceptance; M = Present moment awareness; SACX = Self-as-context; D = Defusion; V = Purpose and values; CA = Committed action; EA = Experiential avoidance; LM = Lack of present moment awareness; SACN = Self-as-content; F = Fusion; LV = Lack of contact with values; and I = Inaction.

baseline, the delta increases (albeit not in a linear way). But after a certain threshold (beyond the value 4.00), the results decline after the workshop. For Group 2, we continue to see increases, but the increase becomes smaller the higher the baseline, with the exception of the last category (beyond the value of 4.00). For EI, Group 1's participants with a low baseline (between 1.50 and 1.99) still increase in average value of EI. After that value, a linear decrease in EI can be seen. This pattern is the same for Group 2. For GSE, Group 1 reveals a linear decline when the baseline increases. The result for Group 2 is curvilinear. The average deltas for all sub-processes for EI and EF for both Group 1 and 2 are summarized in Table 9.

Looking at this set of exploratory results, we conclude that there might be an interaction effect, perhaps even a curvilinear effect, of the delta with the baseline value of the respective constructs (i.e., GSE, EF, and EI). The lower the starting baseline value before the workshop,

**Table 8. Improvement vis-à-vis baseline results.**

| EMOTIONAL FLEXIBILITY GROUP 1 difference versus baseline (*n* = 21) | | | | | EMOTIONAL FLEXIBILITY GROUP 2 difference versus baseline (*n* = 17) | | | | |
|---|---|---|---|---|---|---|---|---|---|
| Between Lower | Between Upper | Category | *n* | Average Delta | Between Lower | Between Upper | Category | *n* | Average Delta |
| 2.50 | 2.99 | a | 2 | 1.62 | 2.00 | 2.49 | a | 2 | 1.75 |
| 3.00 | 3.49 | b | 2 | 0.28 | 2.50 | 2.99 | b | 6 | 0.89 |
| 3.50 | 3.99 | c | 9 | 0.41 | 3.00 | 3.49 | c | 3 | 0.22 |
| 4.00 | 4.49 | d | 5 | -0.21 | 3.50 | 3.99 | d | 4 | 0.17 |
| 4.50 | 4.99 | e | 3 | -0.62 | 4.00 | 4.49 | e | 2 | 0.32 |
| **Overall Baseline Mean** | 3.88 | **Overall Delta** | | 0,22 | **Overall Baseline Mean** | 3.19 | **Overall Delta** | | 0.64 |
| EMOTIONAL INFLEXIBILITY GROUP 1 difference versus baseline (*n* = 21) | | | | | EMOTIONAL INFLEXIBILITY GROUP 2 difference versus baseline (*n* = 17) | | | | |
| Between Lower | Between Upper | Category | *n* | Average Delta | Between Lower | Between Upper | Category | *n* | Average Delta |
| 1.50 | 1.99 | a | 3 | 0.21 | 2.00 | 2.49 | a | 5 | 0.14 |
| 2.00 | 2.49 | b | 4 | -0.01 | 2.50 | 2.99 | b | 5 | -0.22 |
| 2.50 | 2.99 | c | 9 | -0.44 | 3.00 | 3.49 | c | 4 | -0.55 |
| 3.00 | 3.49 | d | 3 | -0.61 | 3.50 | 3.99 | d | 2 | -1.20 |
| 3.50 | 3.99 | e | 0 | n.a. | 4.00 | 4.49 | e | 1 | -1.77 |
| 4.00 | 4.49 | f | 1 | -1.44 | | | | | |
| 4.50 | 4.99 | g | 1 | -1.86 | | | | | |
| **Overall Baseline Mean** | 2.69 | **Overall Delta** | | -0.40 | **Overall Baseline Mean** | 2.92 | **Overall Delta** | | -0.40 |
| GENERAL SELF EFFICACY GROUP 1 difference versus baseline (*n* = 21) | | | | | GENERAL SELF EFFICACY GROUP 2 difference versus baseline (*n* = 17) | | | | |
| Between Lower | Between Upper | Category | N | Average Delta | Between Lower | Between Upper | Category | N | Average Delta |
| 2.50 | 2.99 | a | 7 | 0.21 | 2.50 | 2.99 | A | 6 | 0.38 |
| 3.00 | 3.49 | b | 9 | 0.09 | 3.00 | 3.49 | B | 8 | 0.06 |
| 3.50 | 3.99 | c | 5 | 0.02 | 3.50 | 3.99 | C | 3 | 0.10 |
| **Overall Baseline Mean** | 3.20 | **Overall Delta** | | 0.10 | **Overall Baseline Mean** | 3.11 | **Overall Delta** | | 0.18 |

**Table 9. Average deltas for sub-processes of EI and EF for Group 1 and 2.**

| | Group 1 | | | Group 2 | | |
|---|---|---|---|---|---|---|
| Dimension/Sub-process | Mean | Delta | % Delta | Mean | Delta | % Delta |
| Emotional Flexibility | 3.88 | 0.22 | 6% | 3.19 | 0.64 | **20%**\*\*\* |
| Acceptance | 3.59 | 0.36 | 10% | 2.69 | 0.74 | **28%**\*\*\* |
| Present moment awareness | 3.96 | 0.24 | 6% | 3.04 | 0.94 | **31%**\*\*\* |
| Self-as-context | 4.20 | 0.03 | 1% | 3.58 | 0.61 | **17%**\*\* |
| Defusion | 3.35 | 0.48 | **14%**\* | 3.02 | 0.69 | **23%**\*\*\* |
| Purpose and values | 4.01 | 0.18 | 5% | 3.39 | 0.39 | **11%**\* |
| Committed Action | 4.15 | 0.04 | 1% | 3.44 | 0.44 | **13%**\*\* |
| Emotional Inflexibility | 2.69 | -0.40 | **-15%**\*\*\* | 2.92 | -0.40 | **-14%**\*\* |
| Experiential Avoidance | 3.17 | -0.42 | -13%\* | 3.25 | -0.04 | -1% |
| Lack of present moment awareness | 2.47 | -0.30 | -12% | 3.13 | -0.76 | **-24%**\*\* |
| Self-as-content | 2.26 | -0.36 | **-16%**\*\* | 2.52 | -0.56 | **-22%**\* |
| Fusion | 2.67 | -0.37 | **-14%**\* | 2.94 | -0.47 | **-16%**\* |
| Lack of contact with values | 2.94 | -0.34 | -12% | 3.32 | -0.34 | -10% |
| Inaction | 2.64 | -0.63 | **-24%**\*\*\* | 2.39 | -0.21 | -9% |

\* Significant at 0.1 level

\*\* Significant at 0.05 level; and

\*\*\* Significant at 0.01 level–all two-tailed.

the higher the increase of the value afterwards (or, in the case of inflexibility, the other way around). However, with a higher baseline, the workshops may add decreasing additional value. Overall, a similar pattern was found for the underlying sub-processes of EF and EI (Table 9). These exploratory results indicate that participants with lower GSE and EF and higher EI to start with benefit most from targeted training in these areas. These results are in support of Hypothesis H2c, but again further research on a larger sample is needed to see if these results can be replicated.

## Discussion

This study is the first to investigate the development of emotional flexibility skills through training activities amongst knowledge workers. We found that these skills can be trained effectively indeed. In so doing, we replicate what others found in a different type of working context [23]. For the measurement of emotional (in)flexibility, moreover, we are the first to use a more fine-grained measurement instrument that captures both emotional flexibility as well as emotional inflexibility, including measures of the distinctive sub-processes (six factors for emotional flexibility, and six for emotional inflexibility). A number of new insights emerged from our study.

First, the measurement instrument held up very well in this new context, even with our low number of responses. The original factor structure was almost perfectly replicated, and the internal validity of the sub-scales measuring the sub-processes is well above the threshold. The first and only validation of the instrument was first published in 2016 [10]. The fact that we replicate the original factorial structure with such high reliabilities in our highly educated professional working context with a relatively small number of responses ($n$ = 59) provides further strong validation, implying additional support for its further use in field settings.

Second, using this fine-grained instrument provides the opportunity to examine the results at the more detailed level of emotional flexibility vis-à-vis inflexibility, as well as all six or twelve sub-processes. Related to this, we found that the development of the emotional (in)flexibility skills moved into meaningful directions in the treatment (training) group, where this was not the case for the control (non-training) group. This indicates that the training activities indeed contribute to clarifying the underlying concepts, and to influence progress in a meaningful way.

Third, we were also able to tentatively examine the baseline learning impact opportunity from both the emotional flexibility and inflexibility perspective, and found exploratory evidence that training activities may have a curvilinear effect on outcomes, depending on baselines at the start of the training. Specifically, those participants with the least favorite scores on all sub-processes may benefit the most from training, but this benefit might well decrease when such scores are higher.

Fourth, and finally, by adding General self-efficacy as an extra outcome variable in our design, we were able to see how an important construct from the perspective of workplace effectiveness was impacted in a positive or negative way by the development of emotional flexibility (or the reduction of emotional inflexibility). Emotional flexibility seems to improve General self-efficacy, again dependent on the pre-training baseline of both emotional flexibility and inflexibility.

### Limitations and future research

In the high-powered working environment of highly educated knowledge professional, we had to accept substantial attrition of participation during the training and measurement phases of our study's design, yet less so for survey participation. Overall, we expect our attrition rates to

be relatively 'normal' for the unique setting we worked in. Although we have tracked progress over a non-trivial period of time (about eight weeks), it may be interesting to see whether or not the impact of the training activities holds over a longer period of time as well. Hence, more research in a similar environment with a larger sample size and impact tracking over a longer period of time will contribute to examining the robustness of the insights and results reported in this paper, also to learn more about the fine-grained performance effects at the level of the twelve sub-processes associated with emotional (in)flexibility.

In this study, we worked with a set-up similar to the waiting list control group design, with the adjustment that the control group was assured of the training after the first series of experimental group training activities were done. During the phase in which the first group went through the training sessions, the control group received no active intervention. This set-up was as much as possible done in a voluntary way, and both groups were highly comparable. All participants could indicate their preference for the timing of the intervention, and that response was treated on a 'first-come-first-serve' basis. Moreover, they could try to exchange their place with someone already confirmed. The opportunity to join was given to everyone in the defined target audience, which means that the assignment to groups was not done randomly. In the setting of the working context, it was considered unethical (or even manipulative) to pick and choose, and assign randomly. With this approach, however, the participants who decided to join likely had more affinity with the topic and/or need for the training, or had perhaps a different level of EF to push back on work demands, which may have influenced the results as we explained. This must be examined in future work.

## Practical implications

The practical implications of this research are multiple. First insights from this study confirm the feasibility of emotional flexibility skills development in the working context, in particular targeted at knowledge workers. Prior work revealed that these skills are highly relevant in the treatment and prevention of mental health challenges. Making these skills training activities part of the standard curriculum for employees can provide huge benefits to organizations that are increasingly experiencing the need for prevention and response to these challenges. Furthermore, the development of emotional flexibility skills gives a boost to the general self-efficacy of knowledge workers, which is related to better performance and well-being [37]. Overall, this study contributes by offering insights into the opportunity and feasibility for organizations to respond to the increasing need to invest in their employees' mental health.

## Supporting information

**S1 Fig.**
(TIF)

**S2 Fig.**
(TIF)

**S1 Appendix.**
(DOCX)

**S2 Appendix.**
(DOCX)

**S3 Appendix.**
(DOCX)

**S1 Raw data.**
(XLSX)

## Author Contributions

**Conceptualization:** Jacqueline Brassey, Csaba Huszka.

**Data curation:** Jacqueline Brassey.

**Formal analysis:** Jacqueline Brassey.

**Investigation:** Jacqueline Brassey, Csaba Huszka.

**Methodology:** Jacqueline Brassey, Csaba Huszka.

**Project administration:** Jacqueline Brassey.

**Resources:** Jacqueline Brassey, Tobias Silberzahn.

**Software:** Jacqueline Brassey.

**Supervision:** Arjen van Witteloostuijn, Nick van Dam.

**Validation:** Jacqueline Brassey.

**Visualization:** Jacqueline Brassey.

**Writing – original draft:** Jacqueline Brassey.

**Writing – review & editing:** Jacqueline Brassey, Arjen van Witteloostuijn.

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
