## [Decision Letter · Decision Letter 0]

8 Jul 2020

PONE-D-20-11080

Emotional Flexibility and General Self-Efficacy: A training Intervention Study with Knowledge Workers

PLOS ONE

Dear Dr. Brassey,

Thank you for submitting your manuscript to PLOS ONE. After careful consideration, we feel that it has merit but does not fully meet PLOS ONE’s publication criteria as it currently stands. Therefore, we invite you to submit a revised version of the manuscript that addresses the points raised during the review process.

While there is certainly merit in this submission, the reviewers point out various concerns that need to be addressed.  If the authors feel they can address these concerns satisfactorily, they are invited to submit a revised version of the manuscript, bearing in mind that the revision might be sent back out for review.

We look forward to receiving your revised manuscript.

Kind regards,

Konstantinos V Petrides

Academic Editor

PLOS ONE

Additional Editor Comments:

While there is certainly merit in this submission, the reviewers point out various concerns that need to be addressed. Please fully address the particularly pressing issue regarding the low sample size of the study. If the authors feel they can address these concerns satisfactorily, they are invited to submit a revised version of the manuscript, bearing in mind that the revision may be sent back out for review.

Journal Requirements:

2. Please provide additional details regarding participant consent. In the Methods section, please state why it was not possible to obtain written consent and whether the ethics committee approved this consent procedure described.

"The main author is a part-time employee of the organization where the research was performed, a part-time affiliated researcher at three Universities. The fourth author is an employee of the same organization and invited the research to be done as part of an overall employee well-being program done internally. None of the other authors are part of this organization and were involved as academic researchers."

We note that one or more of the authors are employed by a commercial company: McKinsey & Company Berlin.

3.1. Please provide an amended Funding Statement declaring this commercial affiliation, as well as a statement regarding the Role of Funders in your study. If the funding organization did not play a role in the study design, data collection and analysis, decision to publish, or preparation of the manuscript and only provided financial support in the form of authors' salaries and/or research materials, please review your statements relating to the author contributions, and ensure you have specifically and accurately indicated the role(s) that these authors had in your study. You can update author roles in the Author Contributions section of the online submission form.

3.2. Please also provide an updated Competing Interests Statement declaring this commercial affiliation along with any other relevant declarations relating to employment, consultancy, patents, products in development, or marketed products, etc. 

Reviewers' comments:

Reviewer's Responses to Questions

**Comments to the Author**

1. Is the manuscript technically sound, and do the data support the conclusions?

Reviewer #1: Yes

Reviewer #2: Partly

Reviewer #3: Yes

2. Has the statistical analysis been performed appropriately and rigorously? 

Reviewer #1: Yes

Reviewer #2: No

Reviewer #3: Yes

3. Have the authors made all data underlying the findings in their manuscript fully available?

Reviewer #1: Yes

Reviewer #2: Yes

Reviewer #3: Yes

4. Is the manuscript presented in an intelligible fashion and written in standard English?

Reviewer #1: Yes

Reviewer #2: Yes

Reviewer #3: Yes

5. Review Comments to the Author

Reviewer #1: Important note: This review pertains only to ‘statistical aspects’ of the study and so ‘clinical aspects’ [like medical importance, relevance of the study, ‘clinical significance and implication(s)’ of the whole study, etc.] are to be evaluated [should be assessed] separately/independently. Further please note that any ‘statistical review’ is generally done under the assumption that (such) study specific methodological [as well as execution] issues are perfectly taken care of by the investigator(s). This review is not an exception to that and so does not cover clinical aspects {however, seldom comments are made only if those issues are intimately / scientifically related & intermingle with ‘statistical aspects’ of the study}.

COMMENTS:

Your ABSTRACT is well drafted but assay type. Please note that it is preferable [refer to item 1b of CONSORT checklist 2010: Structured summary of trial design, methods, results, and conclusions] to divide the ABSTRACT with small sections like ‘Objective(s)’, ‘Methods’, ‘Results’, ‘Conclusions’, etc. which is an accepted practice of most good/standard journals [including PLOS]. It will definitely be more informative then, I guess [even if your article type is ‘Research Article’].

Overall, this study is almost faultless and the manuscript of this article is excellently drafted. However, I have few suggestions:

1. Although use of the ‘tools’ is very appropriate, nature of the data yielded by such tools are generally of ‘ordinal’ level of measurement and so use of non-parametric test(s) is desirable {example Wilcoxon’s Signed Rank test instead of paired ‘t’ test or ‘Spearman’s (rank/non-parametric) correlation coefficient’ instead of ‘Pearson’s correlation coefficient’}.

2. Note that oftentimes recoding of ‘Likert’ scale responses, {example, lines 324-328: General self-efficacy (GSE) scale includes ten items.& all ten items are rated on a four-point scale: 1 = ‘Not at all true’; 2 = ‘Hardly true’; 3 = ‘Moderately true’; and 4 = ‘Exactly true’} as follows, will be useful.

Whenever response options ranged from 1=strongly disagree to 4=strongly agree (or from 1=very bad to 3=neither good nor bad to 5=very good), while using a ‘Likert’ scale responses, recoding [like strongly disagree=(-)2, disagree=(-1), neutral=0, agree=(+)1, strongly agree=(+)2] may yield correct and meaningful ‘arithmetic mean’ which is useful not only for comparison but has absolute meaning, in my opinion. Application of any statistical test(s) assume that meaning of entity used (mean, SD, etc) has a particular meaning. Though ‘α’ [alpha] or most other measures of reliability/correlation will remain same, however.

3. Note that highly significant (large) value of ‘Pearson’s correlation coefficient’ alone does not imply cause-effect relationship. There are other certain criteria which are to be considered before making any causal inference(s) [particularly refer to line 480 –-- effect is ‘not in a linear way’ always].

Statistical test usually used to assess significance of Pearson’s ‘Correlation coefficient (r)’ is ‘t’ [where t = { r � [(n-2) / (1-r2)] }for df=n-2, n is sample size] and here Ho is that the population/standard value of ‘r’ is zero. You need r=0.878 to be significant at 5% if n=5 but you need r=0.273 if n=50 & you need only r=0.088 if n=500. ‘P-value’ heavily depends on sample size. Therefore, it is customary to use the absolute value of ‘r’ guidelines for interpreting positive or negative correlations (and not rely on corresponding ‘P’-value).

4. Data of table-5, table-8, and table-9 could have been analysed in other better ways [consult a biostatistician for advice].

5. Discussing/highlighting ‘new insights emerged from this study’ [lines 506-28] is a very good practice/idea.

6. As said in lines 550-52 that “We have not seen any sign of this self-selection in the data”, however, please note you seldom see any sign of self-selection and [should not believe that a similar effect will be achieved in other studies with random assignment] so, this must be examined in future work.

I agree with what is said at the end that “Overall, this study contributes by offering insights into the opportunity and feasibility for organizations to respond to the increasing need to invest in their employees’ mental health.”.

Reviewer #2: Thank you for submitting this work to PLOS ONE. I found this to be a very interesting experimental work in which the authors have tested whether an intervention to develop emotional flexibility will improve an individuals emotional flexibility and general self-efficacy. However, there are a couple of key issues that need to be addressed.

QUESTIONS:

Is the manuscript technically sound, and do the data support the conclusions?

Has the statistical analysis been performed appropriately and rigorously?

The explanation of these two questions can be found in the following comments.

MAJOR ISSUES:

1. There was substantial attrition in this study, as the authors noted, but the reasons given were not sufficient. While in a workplace setting last minute urgent work may have been a primary reason given by the Author's clients, were there any other characteristics of those individuals who continued compared to those who did not that should/could be mentioned? Attrition was substantial and should be addressed further.

2. The substantial attrition has lead to additional issues pertaining to the statistical analyses and the conclusions derived by the authors.

The most concerning issue is sample size. The statistical analyses used with such a small sample size were not sufficient, or appropriate, and do not support the conclusions derived by the authors. The conclusions are overstated, and this data would be best framed as pilot study data.

Running correlations with such a small sample size is problematic. For example, these results could be driven by one or two outliers rather than being a legitimate effect. Additionally, running PCA with such a small sample size (despite the returned results) is potentially inappropriate. The authors have broken down the baseline EF into buckets some of which had just 1 participant - this is not sufficient to be making conclusions relating to improvements from baseline etc. Finally, we should reconsider using these types of statistical analyses for such small sample sizes and use more appropriate analyses with which to form conclusions.

The primary issues with this work suggest that it would benefit from being re-framed as a pilot study, or a technical report in which the authors are providing some evidence for this methodology to be used in a larger scale study. The introduction and discussion read like a technical report with very little background research but more prominently discussing the methodology and the foundations of the methodology.

I would advise the authors to either collect more data, or to re-frame this paper as a pilot study, or technical document.

Reviewer #3: I’ve read with interest and care your paper titled “Emotional Flexibility and General Self-Efficacy: A training Intervention Study with Knowledge Workers”. I commend your choice to focus on emotional flexibility. I do believe that the development of emotional flexibility among knowledge workers is crucial for their well-being, especially nowadays and in the future

In this review, I will describe the few concerns I have with the current manuscript and try and provide some observations that may assist you in strengthening the study and in pursuing this interesting line of research.

1. Please clarify what was the time frame you have used when collecting responses for the 12 dimensions of the MPFI. Was it “on general” or “in the last two weeks” or something else?

2. Please explain in more detail the possible reasons why the scores on ‘Purpose and values’, and ‘No purpose and values have changed for the control group

In sum, this is an interesting paper. I wish you good luck as you continue working in this area and hope that my few comments are taken in the positive and constructive manner in which they are intended.

6. PLOS authors have the option to publish the peer review history of their article (what does this mean?). If published, this will include your full peer review and any attached files.

Reviewer #1: **Yes: **Dr. Sanjeev Sarmukaddam

Reviewer #2: No

Reviewer #3: **Yes: **Leonidas A. Zampetakis

Assistant Professor

Department of Psychology

University of Crete, Greece

---

## [Author Response · Author response to Decision Letter 0]

31 Jul 2020

Dear dr. Petrides,

Thank you very much for offering the opportunity to revise our paper “Emotional Flexibility and General Self-Efficacy. A Pilot Training Intervention Study with Knowledge Workers” for PLOS ONE. Whilst producing a revised draft, we benefitted greatly from the constructive feedback of all reviewers. We believe we have been able to produce a substantially improved manuscript. We hope you do agree. 

Warmly, 

Jacqueline Brassey

Arjen van Witteloostuijn

Csaba Huszka

Tobias Silberzahn

Nick van Dam

---

## [Editor Report · Decision Letter 1]

4 Aug 2020

Emotional Flexibility and General Self-Efficacy: A Pilot Training Intervention Study with Knowledge Workers

PONE-D-20-11080R1

Dear Dr. Brassey,

We’re pleased to inform you that your manuscript has been judged scientifically suitable for publication and will be formally accepted for publication once it meets all outstanding technical requirements.

Kind regards,

Konstantinos V Petrides

Academic Editor

PLOS ONE

Additional Editor Comments (optional):

Thank you for making the requested changes to your manuscript. I am now pleased to recommend it for publication in PLOS ONE and I look forward to seeing it published in the Journal in due course.

Yours sincerely

K V Petrides

---

## [Editor Report · Acceptance letter]

17 Sep 2020

PONE-D-20-11080R1 

Emotional Flexibility and General Self-Efficacy: A Pilot Training Intervention Study with Knowledge Workers 

Dear Dr. Brassey :

I'm pleased to inform you that your manuscript has been deemed suitable for publication in PLOS ONE. Congratulations! Your manuscript is now with our production department. 

Kind regards, 

on behalf of

Professor Konstantinos V Petrides 

Academic Editor

PLOS ONE